# Oscillations Make Neural Networks Robust to Quantization

## Abstract

We challenge the prevailing view that oscillations in Quantization Aware Training (QAT) are merely undesirable artifacts caused by the Straight-Through Estimator (STE). Through theoretical analysis of QAT in linear models, we demonstrate that the gradient of the loss function can be decomposed into two terms: the original full-precision loss and a term that causes quantization oscillations. Based on these insights, we propose a novel regularization method that induces oscillations to improve quantization robustness. Contrary to traditional methods that focuses on minimizing the effects of oscillations, our approach leverages the beneficial aspects of weight oscillations to preserve model performance under quantization. Our empirical results on ResNet-18 and Tiny ViT demonstrate that this counter-intuitive strategy matches QAT accuracy at $\geq$ 3-bit weight quantization, while maintaining close to full precision accuracy at bits greater than the target bit. Our work therefore provides a new perspective on model preparation for quantization, particularly for finding weights that are robust to changes in the bit of the quantizer – an area where current methods struggle to match the accuracy of QAT at specific bits.

## 1. Introduction

Quantization is the mapping of continuous values to discrete values. In neural networks, quantization reduces the computational complexity and memory requirements by representing weights and/or activations with fewer bits (Gupta et al., 2015). In the case of weight only quantization, this means applying a quantizer $q(\cdot)$ to the network's weights $\mathbf{w}$, with an additional implicit goal of maintaining the original performance i.e. $\mathcal{L}(q(\mathbf{w})) \approx \mathcal{L}(\mathbf{w})$, where $\mathcal{L}(\cdot)$ is a loss function.

---

[1]Anonymous Institution, Anonymous City, Anonymous Region, Anonymous Country. Correspondence to: Anonymous Author <anon.email@domain.com>.

Preliminary work. Under review by the International Conference on Machine Learning (ICML). Do not distribute.

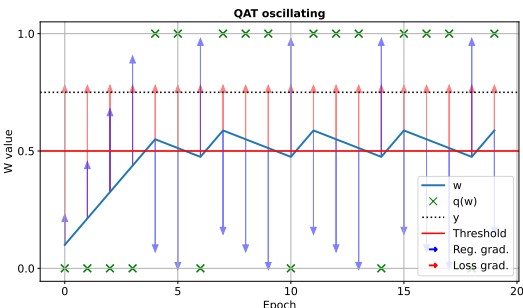

*Figure 1.* Oscillatory behavior in Quantization-Aware Training (QAT) for a simple linear model. The figure shows a quantized linear model $f(x) = q(w)x$ with a single weight $w$, where $x = 1$ and target output $y = 0.75$. When doing squared loss with QAT an additional term is introduced to the gradient (Eq. 14), which causes w to oscillate around the quantization threshold. This oscillation results in $q(w)$ alternating between the 0 and 1 quantization bins.

When training neural networks intended for quantization, an essential step during optimization is accounting for the effects of applying a quantizer to the weights. Quantization introduces a perturbation to the weights. For uniform quantizers, this is bounded by $\frac{s}{2}$, where $s$ is the scale factor. At higher bit widths ($\geq$ 8 bits), this perturbation is small, and standard training procedures often yield weights that are resilient to quantization noise (Nagel et al., 2021). In such cases, applying quantization after training, known as Post-Training Quantization (PTQ), is sufficient to maintain acceptable performance levels (Nagel et al., 2021).

However, as we reduce the bit width to lower precision ($\leq$ 4 bits), the quantization perturbation becomes significant, and the model's performance tends to degrade substantially after quantization. This is because the increased perturbation can lead to larger discrepancy between $q(\mathbf{w})$ and $\mathbf{w}$. To address this challenge, much research has gone into finding strategies to mitigate the effects of quantization on model accuracy, ensuring that the network remains accurate even after low-bit quantization.

Though many methods have been proposed for mitigating the accuracy degradation due to quantization, Quantization-Aware Training (QAT) (Jacob et al., 2018) remains one of the most widely adopted approaches. QAT works by in-

corporating quantization effects directly into the training process - quantizing weights in the forward pass while using the Straight-Through Estimator (STE) (Bengio et al., 2013) for gradient approximation during backpropagation. Research has identified an interesting phenomenon in QAT with the STE, known as weight oscillations, where the quantized weights alternate between two adjacent quantized states during training (Défossez et al., 2021; Nagel et al., 2022). While traditionally viewed as a detrimental effect that should be suppressed through dampening or weight freezing techniques, there also exists evidence suggesting these oscillations might play a more nuanced role in the training dynamics of QAT.

We claim that weight oscillations during training are beneficial and that indeed they are the driving mechanism behind QAT. Our primary contributions that support this claim are:

1. we isolate the mechanism that leads to weight oscillations during QAT (Sec. 4);

2. we develop a regularization method that induces weight oscillations during training using this mechanism (Sec. 5);

3. we show experimentally that weight oscillations are sufficient for preserving performance after quantization on small-scale computer vision tasks (Sec. 6).

Since previous results have shown that weights oscillations are also necessary for good quantization performance with QAT (see Sec. 7 for details), and extrapolating from our experiments, our results suggest that weight oscillations capture all the beneficial effects of QAT while avoiding unintended side-effects. For instance, in our experiments our method avoids overfitting to the bit-width used during training, resulting in superior cross-quantization performance compared to QAT.

## 2. Related Work

The most used strategy to minimize the impact of quantization on model accuracy is to minimize the quantization error. This can be achieved by adjusting the granularity of the quantizer—for instance, using per-channel (Nagel et al., 2019) or block-wise quantization (Dettmers et al., 2022) instead of per-tensor quantization. While these methods reduce quantization error without additional training, they come with increased storage requirements due to extra quantization parameters and may still fall short at very low bit widths, necessitating the combination with other approaches.

Consequently, extensive research has been dedicated to developing techniques that explicitly minimize the quantization error during optimization (Hung et al., 2015; Hi-rose et al., 2017; Li et al., 2019; Choi et al., 2020; Han et al., 2021; Zhong et al., 2025). Given a model $\mathbf{w}$ the hope is that by ensuring $q(\mathbf{w}) \approx \mathbf{w}$, we likely also have $\mathcal{L}(q(\mathbf{w})) \approx \mathcal{L}(\mathbf{w})$, thereby preserving model accuracy after quantization.

An alternative and less explored approach involves training models to be robust to quantization perturbations without necessarily minimizing the quantization error itself. This means finding weights $\mathbf{w}$ such that $\mathcal{L}(q(\mathbf{w})) \approx \mathcal{L}(\mathbf{w})$ even if $q(\mathbf{w})$ is not close to $\mathbf{w}$ (Alizadeh et al., 2020; Chmiel et al., 2020). Such methods focus on enhancing the robustness of the model to the quantization error, leading to better performance at bits different than the ones used in the quantizer, which we will refer to as cross-bit quantization.

A third approach is to train supernets on the desired configurations of the quantizers (Xu et al.; 2023). This approach increases the training complexity and cost, which is not incurred by explicit regularization.

Despite these efforts, the aforementioned strategies often fall short of the accuracy obtained with QAT (Jacob et al., 2018) at individual bits or indirectly rely upon QAT themselves. In short, QAT integrates the quantization process into the training loop allowing the model to adapt to the quantization effects directly. This is done by quantizing the weights during the forward pass and using techniques like the Straight-Through-Estimator (STE) to approximate the gradient of the quantizer (Which has a derivative of zero almost everywhere) during backpropagation (Bengio et al., 2013).

Yet, there is limited understanding of how QAT affects model optimization and why it outperforms other methods. One phenomenon observed during QAT is weight oscillations (Défossez et al., 2021; Nagel et al., 2022), which are periodic changes in the value of the quantized weight between two adjacent quantization levels. It is speculated in these works that that the abrupt changes in values caused by oscillations can interfere negatively with optimization. Oscillations are assumed to be undesirable side effects caused by the use of the STE during backpropagation, as the STE allows gradients to pass through the rounding operation in the quantizer, which has a gradient of zero almost everywhere (Défossez et al., 2021; Nagel et al., 2022).

Several approaches have been suggested to mitigate oscillations, such as dampening or freezing the oscillating weights, which have shown improved accuracy (Nagel et al., 2022; Gupta & Asthana, 2024). However, the reported gains are sometimes marginal, and these methods may inadvertently also hinder the optimization process. For instance, Nagel et al. (2022) notes that freezing or dampening weights too early during training can hurt optimization, indicating that oscillations might contribute to finding better quantized min-

ima. Liu et al. (2023) propose that weights with low oscillation frequency should be frozen, where as high-frequency ones should be left unfrozen, under the rational that high frequency means the network has little confidence in what value to quantize the weight to, where as low frequency means the optimal weight lies close to a quantization level.

Though QAT often provides the best accuracy for a given target bit, degradation to a lesser or greater extent exists when the bit of the quantizer is different to the one seen during training, ie. cross-bit quantization (Alizadeh et al., 2020; Chmiel et al., 2020). This means QAT requires training and storing of weights for each desired bit width. This specialization can also pose challenges when deploying models across different hardware platforms, each potentially using different quantization schemes (Reddi et al., 2020), making it difficult to develop models which can be easily quantized at deployment according to end-user requirements.

This makes robust quantization methods an interesting research avenue, especially if they could be improved to match the individual bit performance of QAT. In this work, we aim to deepen the understanding of how QAT influences model optimization, particularly focusing on the role of weight oscillations and their relation to robustness.

## 3. Preliminaries

### 3.1. Quantization

A quantizer divides a continuous input range into quantization bins, where each bin is represented by a specific quantization level. The boundaries between bins are called quantization thresholds. During quantization, any value within a bin is mapped to that bin's quantization level. With a uniform quantizer, the step size (the distance between two adjacent quantization levels) is equal to the scale factor $s$.

We consider a uniform symmetric quantizer with a max-range scale factor. The quantization operation $q(\cdot)$ can then be expressed as

$$q(\mathbf{w}) = s \cdot \left\lceil \frac{\mathbf{w}}{s} \right\rfloor \tag{1}$$

Here, $s$ represents the scale factor and $\lceil \cdot \rfloor$ denotes the rounding operation.

The scale factor $s$ is set to cover the range of $\mathbf{w}$ as this removes the need for the usual clamping operation in the quantizer, while keeping the number of bins symmetric around 0:

$$s = \frac{\max(|\alpha|, |\beta|)}{2^{b-1} - 1} \tag{2}$$

Where $b$ is the bit in the quantizer and $\alpha, \beta$ are the min. and max. values respectively of the layer wise weight $\mathbf{w}$.

The quantization process introduces quantization error $\boldsymbol{\Delta}$, defined as the difference between the original and quantized values:

$$\boldsymbol{\Delta}(\mathbf{w}) = \mathbf{w} - q(\mathbf{w}) \tag{3}$$

Due to the uniform quantizer, for all bins the absolute error is bounded between $0 \le |\boldsymbol{\Delta}| \le s/2$, which is maximized at quantization thresholds and 0 at quantization levels.

### 3.2. Quantization-Aware Training

While there exist many variants of QAT, fundamentally the forward pass is performed using the quantized weights $q(\mathbf{w})$ in most variants of QAT (Jacob et al., 2018; Krishnamoorthi, 2018), simulating the effect of using low-precision weights. In principle the gradient for the weights during QAT is given by:

$$\frac{\partial \mathcal{L}(q(\mathbf{w}))}{\partial \mathbf{w}} = \frac{\partial \mathcal{L}(q(\mathbf{w}))}{\partial q(\mathbf{w})} \cdot \frac{\partial q(\mathbf{w})}{\partial \mathbf{w}} \tag{4}$$

A problem with the above formulation is that the gradient of the rounding operation used in the quantizer is zero almost everywhere, causing the last term to interrupt gradient-based learning. A popular solution to this problem is to use the so-called Straight-Through Estimator (STE) (Bengio et al., 2013). We define the STE to be the operator $\frac{\hat{\partial}}{\partial \mathbf{x}}$ such that $\frac{\hat{\partial} f}{\hat{\partial} x}$ is obtained by computing $\frac{\partial f}{\partial x}$ and in the resulting expression replacing $q'$ (the derivative of $q$) by the constant function equal to 1. In other words, if $\frac{\partial f}{\partial x} = g(\ldots, q', \ldots)$ then $\frac{\hat{\partial} f}{\hat{\partial} x} = g(\ldots, 1, \ldots)$.

## 4. Oscillations in QAT

Previous studies have explored linear models to analyze the behavior of QAT and the phenomenon of weight oscillations (Défossez et al., 2021; Nagel et al., 2022; Liu et al., 2023; Gupta & Asthana, 2024). Inspired by these works, we analyze a linear regression model to gain theoretical insights into the optimization dynamics during QAT.

Consider a linear model with a single weight $w$, input $x$ and target $y \in \mathbb{R}$. The quantized version of this model is defined as $f(x) = q(w)x$, where $q(\cdot)$ is the quantizer from Eq. 1. The quadratic loss for the quantized model is given by

$$\mathcal{L}(q(w)) = \frac{1}{2}(q(w)x - y)^2. \tag{5}$$

Our goal in this section is to understand how QAT affects the full precision optimization process. For a given loss function $\mathcal{L}(\cdot)$ with quantized weights, we have

$$\mathcal{L}(q(w)) = \mathcal{L}(w) + \mathcal{L}(q(w)) - \mathcal{L}(w) \tag{6}$$

We can then expand the difference caused by quantization as follows

$$\delta_{\mathcal{L}} = \mathcal{L}(q(w)) - \mathcal{L}(w) \tag{7}$$

$$= \frac{1}{2}\left((q(w)x - y)^2 - (wx - y)^2\right) \tag{8}$$

$$= \frac{1}{2}\left((q(w)x)^2 - (wx)^2 - 2y(q(w)x - wx)\right) \tag{9}$$

$$= \frac{1}{2}\left(x^2\left(q(w)^2 - w^2\right)\right) + (yx(w - q(w))) \tag{10}$$

This expression decomposes the loss difference into a quadratic term $\frac{1}{2}x^2(q(w)^2 - w^2)$ and a linear term $yx(w - q(w))$.

Next we derive the gradient of $\delta_{\mathcal{L}}$ wrt. $w$:

$$\frac{\partial \delta_{\mathcal{L}}}{\partial w} = \frac{\partial}{\partial w}\left(\mathcal{L}(q(w)) - \mathcal{L}(w)\right) \tag{11}$$

$$= \frac{\partial}{\partial w}\left(\frac{1}{2}x^2(q(w)^2 - w^2) + yx(w - q(w))\right) \tag{12}$$

$$= x^2\left(q(w)\frac{\partial q(w)}{\partial w} - w\right) + yx\left(1 - \frac{\partial q(w)}{\partial w}\right) \tag{13}$$

Using the STE and recalling that $\frac{\hat{\partial} q}{\hat{\partial} w} = 1$ the expression of the STE gradient simplifies to

$$\frac{\hat{\partial} \delta_{\mathcal{L}}}{\hat{\partial} w} = x^2(q(w) - w) = -x^2\mathbf{\Delta}(w). \tag{14}$$

To see how this gives rise to oscillations, for an arbitrary $w$, denote $w_0$ the upper discretization threshold $w_0 = q(w) + s/2$. For $\varepsilon \in (0, s/2)$ note that we have $q(w_0 - \varepsilon) = q(w)$ and $q(w_0 + \varepsilon) = q(w) + s$ so that

$$\mathbf{\Delta}(w_0 + \varepsilon) = q(w) + s/2 + \varepsilon - (q(w) + s) \tag{15}$$

$$= -s/2 + \varepsilon, \tag{16}$$

$$\mathbf{\Delta}(w_0 - \varepsilon) = q(w) + s/2 - \varepsilon - q(w) \tag{17}$$

$$= s/2 - \varepsilon. \tag{18}$$

Assuming $x \neq 0$, the negative STE gradient "flips" from $-s/2$ to $s/2$ as the weight $w$ passes the quantization threshold $w_0$ from above, pushing the weight back towards the threshold. We note that the STE gradient is 0 at the special value $w = q(w)$, but the preceding argument shows that this is an unstable critical point and gradient noise will immediately cause the weights to move away from it. When combined with (stochastic) gradient descent and a finite discretization timestep we can identify this as the driving mechanism behind oscillations during training with QAT (Fig. 1).

We can also see how the dynamics lead to clustering around quantization thresholds by looking at the sign of $\mathbf{\Delta}$ for different values of $w$. For a weight $w$ let $d_{\text{low}}(w)$ and $d_{\text{up}}(w)$ denote the distance from $w$ to the upper and lower thresholds, $d_{\text{low}}(w) = w - \left(q(w) - \frac{s}{2}\right) = \mathbf{\Delta}(w) + \frac{s}{2}$ and $d_{\text{up}}(w) = \left(q(w) + \frac{s}{2}\right) - w = \frac{s}{2} - \mathbf{\Delta}(w)$ respectively. If $w$ is closest to the upper threshold we have

$$d_{\text{up}} < d_{\text{low}} \implies \tfrac{s}{2} - \mathbf{\Delta} < \mathbf{\Delta} + \tfrac{s}{2} \implies \mathbf{\Delta} > 0 \tag{19}$$

While if $w$ is closest to the lower threshold

$$d_{\text{low}} < d_{\text{up}} \implies \mathbf{\Delta} + \tfrac{s}{2} < \tfrac{s}{2} - \mathbf{\Delta} \implies \mathbf{\Delta} < 0 \tag{20}$$

We emphasize that this mechanism causes the weights to move towards the quantization thresholds (the edges of quantization "bins") as opposed to the quantization levels (the centers of the quantization "bins").

## 5. Regularization Method

Based on our theoretical observations in the one weight linear model, we now investigate empirically if the mechanism in Eq. (14) is sufficient to introduce weight oscillations in neural networks.

From the quantization difference in Eq. 10 and the STE gradient derived in Eq. 14, we have:

$$\frac{\partial \mathcal{L}(q(w))}{\partial w} = \frac{\partial \mathcal{L}(w)}{\partial w} - x^2\mathbf{\Delta}(w) \tag{21}$$

where the first term is the gradient of the original full-precision loss, and the second term causes the quantization oscillations in QAT.

In order to emulate the effects of QAT, we propose a regularization term so that the training objective becomes:

$$\mathcal{L}(q(\mathbf{w})) = \mathcal{L}(\mathbf{w}) + \mathcal{R}_\lambda(\mathbf{w}) \tag{22}$$

where we let the regularization term be similar to the quadratic term in Eq. (10):

$$\mathcal{R}_\lambda(\mathbf{w}) = \frac{\lambda}{2}\sum_\ell \frac{1}{n_\ell}\sum_{i=1}^{n_\ell}\left(q(w_i^\ell)^2 - (w_i^\ell)^2\right). \tag{23}$$

Here $\lambda \geq 0$ is a hyperparameter that controls the amount of regularization, $\ell$ ranges over the layers in the model and $i$ over the weights in each layer.

Using the STE, $\frac{\hat{\partial} q}{\hat{\partial} \mathbf{w}} = 1$, we have the following expression for the gradient:

$$\frac{\hat{\partial}}{\hat{\partial} w_i^\ell}\mathcal{R}_\lambda(\mathbf{w}) = \frac{\lambda}{n_\ell}\left(q(w_i^\ell) - w_i^\ell\right) = -\frac{\lambda}{n_\ell}\mathbf{\Delta}(w_i^\ell). \tag{24}$$

By the same reasoning as in Sec. 4 this pulls the weight $w_i^\ell$ towards the quantization threshold and causes the gradient to "flip" as $w_i^\ell$ crosses the threshold. We expect this to lead to oscillations based on the same mechanism as in the model from Sec. 4.

Figures 2 and 3 show the results of an experiment where we observe the weight distributions, and measured the oscillations, during training of a neural network (ResNet-18) with varying degrees of regularization, respectively. For comparison purposes the figures also shows the weight distributions and oscillations observed during training with QAT. Using the definition of an oscillation established in Nagel et al. (2022), we count an oscillation at epoch $i > 1$ if $q(w_t) \neq q(w_{t-1})$ and the direction of the change in the quantized space is opposite to that of the previous change. We note though that this method of counting misses the first threshold crossing during an oscillation A.5

Our first observation is that QAT displays more oscillations – also seen as clustering around the quantization threshold in Fig. 2-a) – than a baseline model without QAT or regularization (corresponding to $\lambda = 0$ in Fig. 3-b)) . As we increase $\lambda$ we observe that the number of oscillations as well as the clustering increases. This confirms that our regularizer can indeed induce oscillations similar to QAT during the training of deep neural networks. At $\lambda = 1$ (Fig. 3-c)) the number of oscillations observed with our regularizer is similar to the behaviour of QAT, lending support to our hypothesis that the mechanism in (15) is indeed at the root of the oscillations observed when training neural networks with QAT.

# 6. Experiments & Results

In this section we empirically try to answer the question: is it sufficient to induce weight oscillations during training in order to get the benefits of QAT?

We answer this question mostly affirmatively for ResNet and Vision Transformer architectures, based on the results of training ResNet-18 and Tiny ViT on the CIFAR-10 dataset. This is both in a training-from-scratch setting and when fine-tuning pretrained models. In all our experiments we use the regularizer $\mathcal{R}_\lambda$ defined in Eq. (23) to induce oscillations.

In the following subsections we first describe the experimental setup, then we present the accuracy results from training-from-scratch and fine-tuning models trained with different quantization levels for the quantizer in $\mathcal{R}_\lambda$ or QAT and finally, we present the cross-bit accuracy of the fine-tuned models. We train models at ternary (3 possible values: -1, 0, 1), 3-bit and 4-bit. This is in line with contemporary research, where the emphasis lies on quantization at 4-bit and below since the challenges of maintaining accuracy are more significant compared to quantization at higher bit widths.

## 6.1. Experimental setup

We conducted our experiments using the CIFAR-10 dataset (Krizhevsky et al., 2009) without data augmentation. We evaluated three architectures; A multi-layer perceptron with 5 hidden layers and 256 neurons per layer (MLP5), ResNet-18 (He et al., 2016) and Tiny Vision transformer (Tiny ViT) (Wu et al., 2022).

For each architecture we used the Adam optimizer (Kingma, 2014) and tested multiple configurations: A baseline model to establish optimal floating-point accuracy and post-training quantization (PTQ) performance, a model with QAT and a model with our approach. The two latter configurations are trained using a ternary, 3-bit, and 4-bit quantizer.

**Training from Scratch** For the MLP5 architecture, we used a learning rate of $10^{-3}$ and regularization parameter $\lambda$=1. The ResNet-18 was trained with a learning rate of $10^{-3}$ and $\lambda$=0.75 (see Appx. A.2 for our hyperparameter selection). We modified the ResNet-18 architecture by replacing the input layer with a smaller $3 \times 3$ kernel and adapting the final layer for 10-class classification of both ResNet-18 and Tiny ViT. Training proceeded for a maximum of 100 epochs with early stopping triggered after 10 epochs without improvement in validation performance. For quantized models, we monitored the quantized validation accuracy at the target bit precision, while for the baseline, we tracked floating-point accuracy.

**Fine-tuning** We fine-tuned two ImageNet-1k (Deng et al., 2009) pre-trained models on CIFAR-10: a Tiny ViT (learning rate: $10^{-4}$, $\lambda$=1) and a ResNet-18 (learning rate: $10^{-3}$, $\lambda$=1). To maintain compatibility with the pre-trained architectures, we upsampled CIFAR-10 images to $224 \times 224$ pixels. The $\lambda$ parameter selection process for Tiny ViT is detailed in Appx. A.2. Fine-tuning continued for up to 200 epochs, with early stopping after 30 epochs without improvement, using the same accuracy metrics as training from scratch.

**Quantization** We implemented weight quantization using a per-tensor uniform symmetric quantizer as defined in Eq. 1. The quantization range was determined by computing minimum and maximum values per layer. In our implementation of ResNet-18 (11M parameters) all layers except batch normalization were quantized, covering 99.96% of parameters. For Tiny ViT (5.5M parameters) quantization was applied to MLP, Self-Attention, and key-query-value projection layers, encompassing 97.18% of parameters. And lastly for the MLP5 model all layers were quantized.

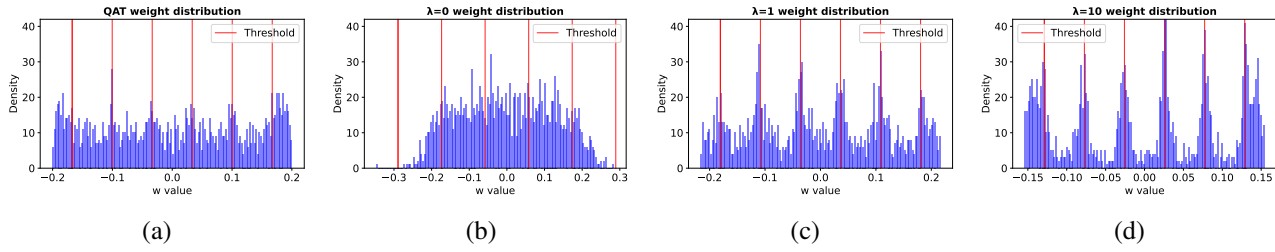

(a)          (b)          (c)          (d)

*Figure 2.* Weight distribution analysis of ResNet-18's first convolutional layer after 50 epochs of training from scratch. a) Weight distribution under QAT with a 3-bit quantizer. b)-d) Our proposed regularization approach with a 3-bit quantizer at varying regularization strengths ($\lambda = 0, 1, 10$, from left to right). When $\lambda = 0$, the training reduces to standard optimization. The QAT distribution (leftmost) exhibits the characteristic threshold clustering behavior. As $\lambda$ increases, we observe progressively stronger clustering of weights around quantization thresholds, illustrating the relationship between regularization strength and weight clustering.

.

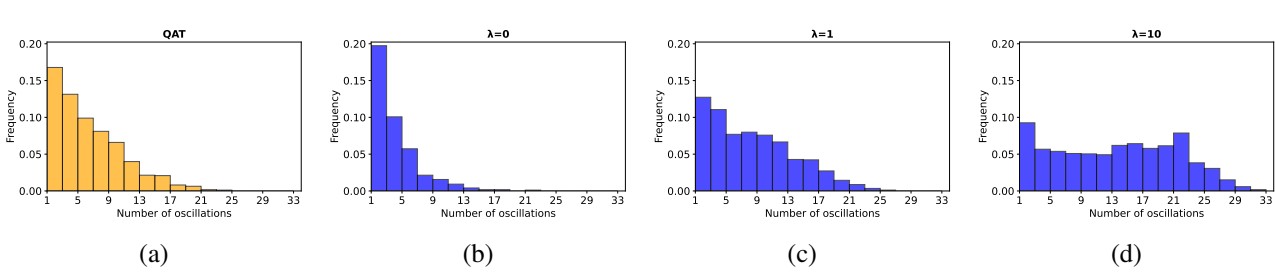

(a)          (b)          (c)          (d)

*Figure 3.* The plots show the distribution of weights with oscillation counts $> 0$ when training with a) QAT and b)-d) our regularizer for different values of $\lambda$. Here $\lambda = 0$ corresponds to a full precision model where our regularizer has no influence on training. The y-axis represents the percentage of total weights in the first convolutional layer of a ResNet-18 trained from scratch for 50 epochs, while the x-axis shows the oscillation count. Following the oscillation definition from (Nagel et al., 2022), we count oscillations at each epoch during training. The results demonstrate that QAT produces a significantly higher proportion of oscillating weights compared to $\lambda = 0$. Furthermore, we observe that as we increase $\lambda$ a greater percentage of weights oscillates.

### 6.2. Training-from-scratch

Table 1 shows the results from training an MLP and ResNet-18 from scratch on the CIFAR-10 dataset. Our regularization method (OsciQuant) demonstrates improvements compared to the PTQ baseline from ternary quantization. More importantly, it also matches the performance of QAT at bit widths of 3 and 4.

For both models we see that at 3-bit and 4-bit, our method exhibits similar performance as QAT but with less variability, while not differing significantly in the average number of training epochs required. With both models, QAT and OsciQuant are competitive with the full-precision baseline, although we observe an increased number of training epochs. Notably, both OsciQuant and QAT significantly outperform PTQ when applied to the full precision baseline.

### 6.3. Fine-tuning

Table 2 summarizes the test accuracies for fine-tuning using our OsciQuant method and QAT on ResNet-18 and Tiny ViT architectures. The observations are roughly in line with the results observed for training from scratch in the previous section with the exception of the number of epochs required

for fine-tuning.

On the ResNet architecture both QAT and our model train for significantly longer than the full precision baseline. As is the case for training from scratch, we see an increase in ternary performance compared to PTQ, but QAT still ourperforms our method in the ternary setting. Our regularization and QAT show comparable performance when quantized at 3 bits and 4 bits, while achieving test accuracy close to the full precision model at 4-bits.

The general trend regarding accuracy is identical for the vision transformer experiments, while we again note the high number of epochs require for both methods when fine-tuning, compared to the full precision baseline.

### 6.4. Robustness to cross-bit quantization

As described above, the goal of our proposed regularization term is to train a model that maintains performance after quantization. Since the regularization term involves a quantization operator, we need to choose the quantization level in the regularization term. In this experiment we evaluated the robustness of our method and QAT towards quantization at levels different from the ones used during training.

| Model | Quantization method | Accuracy | Mean Epochs |
| --- | --- | --- | --- |
| MLP5 | Baseline FP32 | 51.43 ± 0.39 | 14 |
| | Ternary PTQ | 10.00 ± 0.02 | 14 |
| | Ternary QAT | 49.20 ± 1.34 | 24 |
| | Ternary OsciQuant | 36.49 ± 0.51 | 14 |
| | 3-bit PTQ | 20.97 ± 5.64 | 14 |
| | 3-bit QAT | 50.53 ± 1.43 | 33 |
| | 3-bit OsciQuant | 48.48 ± 0.29 | 15 |
| | 4-bit PTQ | 46.50 ± 0.76 | 14 |
| | 4-bit QAT | 51.39 ± 0.60 | 26 |
| | 4-bit OsciQuant | 50.72 ± 0.47 | 19 |
| ResNet-18 | Baseline FP32 | 83.26 ± 1.07 | 24 |
| | Ternary PTQ | 10.00 ± 0.01 | 24 |
| | Ternary QAT | 79.62 ± 6.42 | 42 |
| | Ternary OsciQuant | 61.5 ± 1.82 | 56 |
| | 3-bit PTQ | 77.79 ± 4.0 | 24 |
| | 3-bit QAT | 82.51 ± 2.14 | 37 |
| | 3-bit OsciQuant | 81.77 ± 0.46 | 41 |
| | 4-bit PTQ | 82.11 ± 1.21 | 24 |
| | 4-bit QAT | 82.66 ± 2.57 | 28 |
| | 4-bit OsciQuant | 83.74 ± 0.59 | 32 |

*Table 1.* Comparison of accuracy when training from scratch on CIFAR-10. Results show classification accuracy and mean training epochs for MLP5 and ResNet-18 across different quantization approaches and bit-widths. Results is means and standard deviations over 5 random seeds.

| Model | Quantization method | Accuracy | Mean Epochs |
| --- | --- | --- | --- |
| ResNet-18 | Baseline FP32 | 88.50 ± 0.64 | 4 |
| | Ternary PTQ | 10.01 ± 0.01 | 4 |
| | Ternary QAT | 77.02 ± 7.57 | 47 |
| | Ternary OsciQuant | 44.59 ± 3.30 | 35 |
| | 3-bit PTQ | 10.28 ± 0.48 | 4 |
| | 3-bit QAT | 85.69 ± 1.83 | 25 |
| | 3-bit OsciQuant | 84.94 ± 1.59 | 27 |
| | 4-bit PTQ | 35.56 ± 9.05 | 4 |
| | 4-bit QAT | 87.71 ± 1.14 | 26 |
| | 4-bit OsciQuant | 87.08 ± 0.72 | 24 |
| Tiny ViT | Baseline FP32 | 96.11 ± 0.31 | 6 |
| | Ternary PTQ | 9.39 ± 1.11 | 6 |
| | Ternary QAT | 73.53 ± 0.77 | 140 |
| | Ternary OsciQuant | 13.51 ± 1.32 | 28 |
| | 3-bit PTQ | 11.56 ± 1.99 | 6 |
| | 3-bit QAT | 88.13 ± 0.60 | 131 |
| | 3-bit OsciQuant | 88.68 ± 1.08 | 108 |
| | 4-bit PTQ | 21.57 ± 5.33 | 6 |
| | 4-bit QAT | 94.96 ± 0.33 | 57 |
| | 4-bit OsciQuant | 94.82 ± 0.51 | 90 |

*Table 2.* Comparison of accuracy when fine-tuning on models pre-trained on ImageNet-1k. Results show classification accuracy and mean training epochs for MLP5 and ResNet-18 across different quantization approaches and bit-widths. Results is means and standard deviations over 5 random seeds.

For OsciQuant, we applied a regularization term with the training bit width during training and applied PTQ after training finished at a different quantization level. For QAT we trained using the training bit width and afterwards applied PTQ to the latent weights. For each method we also evaluated the corresponding model without PTQ, directly using the latent weights for inference (reported as FP32).

Table 3 shows the results from the experiment. A first observation is that the models produced by our method consistently achieve nearly full-precision accuracy when quantized at 8-bit or when used without quantization, irrespective of the quantization level used during training. This contrasts with QAT, which produces a viable 8-bit or full-precision model only when trained with at least 4-bit.

Furthermore we see that our method mostly maintains performance when trained at 3 or 4-bit and quantized at bit level of 3 or 4-bit. QAT also achieves this for Tiny ViT but for ResNet, the accuracy of QAT trained at 3-bit and quantized at other bit widths is barely above random guessing.

Regarding training with ternary quantization, we see that our method produces models that achieve near full precision performance for ResNet when quantized at 3-bit or higher. Ternary training for ViT is somewhat peculiar in that it fails to produce a model that is viable when quantized to ternary, whereas the performance of the resulting models starts to show a high level of variability at 4-bit and finally reaches close to full-precision accuracy at 8-bit. In contrast, for both ResNet and ViT, the performance of QAT degrades completely to random guessing when trained with ternary quantization and evaluated at any other quantization level.

## 7. Discussion

We have shown that training with weight oscillations induced via regularization is sufficient in most cases to maintain performance after quantization for ResNet and Tiny ViT. This begs the question whether weight oscillations are also a necessary part of the QAT training process. Indeed, some previous work already points towards this. There are examples claiming that both dampening and/or freezing of oscillations too early in the training process is detrimental to performance after quantization (Nagel et al., 2022; Han et al., 2021). And in other case presented in Liu et al. (2023), freezing only the low frequency oscillating weights improves performance. This suggests that weight oscillations are both a necessary and sufficient part of QAT, at least in the early phases of the training process. This further supports our hypothesis that oscillations in QAT have a positive effect on quantization robustness.

Additionally, there might be further benefits to our regularization approach compared to QAT. Our method aims to isolate this crucial part of the training process. This is arguably a more principled approach compared to QAT, where quantization during training combined with STE can lead to a number of side-effects beyond oscillations, which can be highly non-intuitive. We present a simple example in the Appendix Sec. A.1 where replacing a single scalar weight

| Model | Train bit ↓ / Eval. bit → | FP32 | Ternary | 3-bit | 4-bit | 8-bit |
|---|---|---|---|---|---|---|
| **ResNet-18** | Baseline (PTQ) | 88.50 ± 0.64 | 10.01 ± 0.01 | 10.28 ± 0.48 | 35.56 ± 9.05 | 88.45 ± 0.64 |
| | Ternary QAT | 10.39 ± 0.71 | **77.02 ± 7.57** | 9.75 ± 0.77 | 10.03 ± 0.51 | 10.35 ± 0.63 |
| | Ternary OsciQuant | **87.44 ± 0.56** | 44.59 ± 3.30 | **85.42 ± 1.13** | **87.03 ± 0.65** | **87.42 ± 0.56** |
| | 3-bit QAT | 16.89 ± 4.97 | 10.01 ± 0.04 | 85.69 ± 1.83 | 17.42 ± 4.96 | 16.56 ± 4.32 |
| | 3-bit OsciQuant | **87.86 ± 0.42** | **20.19 ± 10.74** | 84.94 ± 1.59 | **87.56 ± 0.38** | **87.86 ± 0.42** |
| | 4-bit QAT | 87.75 ± 1.13 | 10.13 ± 0.29 | 82.08 ± 6.25 | 87.71 ± 1.14 | 87.76 ± 1.12 |
| | 4-bit OsciQuant | 87.85 ± 0.49 | 11.91 ± 0.87 | 85.57 ± 1.10 | 87.08 ± 0.72 | 87.87 ± 0.49 |
| **Tiny ViT** | Baseline (PTQ) | 96.11 ± 0.31 | 9.39 ± 1.11 | 11.56 ± 1.99 | 21.57 ± 5.33 | 96.03 ± 0.34 |
| | Ternary QAT | 10.62 ± 1.29 | **73.53 ± 0.77** | 11.52 ± 1.82 | 11.13 ± 1.75 | 10.61 ± 1.26 |
| | Ternary OsciQuant | **95.79 ± 0.58** | 13.51 ± 1.32 | 12.53 ± 3.66 | 54.93 ± 27.32 | **95.76 ± 0.59** |
| | 3-bit QAT | 86.94 ± 0.91 | **19.78 ± 6.04** | 88.13 ± 0.60 | 86.69 ± 0.62 | 86.95 ± 0.89 |
| | 3-bit OsciQuant | **96.47 ± 0.11** | 9.48 ± 1.64 | 88.68 ± 1.08 | **95.35 ± 0.18** | **96.50 ± 0.11** |
| | 4-bit QAT | 95.14 ± 0.29 | 11.11 ± 1.84 | 59.86 ± 19.95 | 94.96 ± 0.33 | 95.13 ± 0.28 |
| | 4-bit OsciQuant | **96.54 ± 0.09** | 11.90 ± 1.29 | 70.23 ± 12.75 | 94.82 ± 0.51 | **96.55 ± 0.09** |

*Table 3.* Cross-bit evaluation of pre-trained ImageNet-1k models fine-tuned on CIFAR-10. Grey background is the target-bit accuracy. Models are trained using different quantization methods (QAT and ours) and bit-widths (ternary, 3-bit, and 4-bit), then evaluated across various bit-widths ranging from ternary to FP32. The grey diagonal shows the results for the bit used during training. Results are means and standard deviations over 5 random seeds. All significant differences between QAT and OsciQuant are shown in bold face.

by a product of two scalar weights leads to a non-trivial change in training dynamics when using QAT with the STE.

On the other hand, while it is not clear what the additional effects are during QAT, we do note two consistent deviations from the QAT performance when using our regularization method: QAT outperforms regularization at ternary quantization, whereas our regularization method outperforms QAT in cross-bit accuracy for the ternary and 3-bit case. In A.4, we see how it seems that the cross-bit performance for QAT is upper-bounded by the target-bit performance, which might explain the subpar QAT performance at cross-bit compared to our regularization method which seems bounded by the full precision accuracy. Additionally we can note that while it is stated in Alizadeh et al. (2020); Chmiel et al. (2020) that QAT is not robust to cross-bit quantization, A.4 shows that for some cases the robustness is tied closely to how long the model is trained after the target bit accuracy has converged.

Finally we note in A.2 that in the ResNet-18 model, we see similar results for the hyperparameter sweep for different $\lambda$s, which might suggest that the key for robustness is the presence of oscillations and not their precise nature.

**Limitations** In our experiments we observed that the robustness to cross-bit quantization improves in later training epochs. In order to further improve robustness one might consider an early stopping criterion that evaluates the performance on cross-bit quantization, which was not done in this work. The same approach could also increase cross-bit quantization robustness of QAT although to a lesser degree than for our method.

We performed our experiments on the CIFAR-10 dataset which might make it more difficult to compare our results with other published works that provide benchmark results for other datasets such as ImageNet-1k.

## 8. Conclusion

Based on the analysis of a toy model we proposed the hypothesis that weight oscillations during training in deep neural networks make the model robust to quantization.

In Sections 4 and 5 we explain on a toy model how training with QAT and STE leads to oscillations and propose a regularizer that encourages this oscillating behaviour. We confirm that as we increase the strength of the regularization, we empirically observe the appearance of clustering together with oscillations.

Finally we experimentally confirm that the regularizer indeed leads to consistent robustness towards quantization for quantization levels above ternary. Our regularization method achieves comparable performance to QAT above ternary quantization when quantizing to the target-bit seen during optimizing and shows increased robustness compared to QAT in cross-bit quantization with bits greater than the target-bit used in the quantizer during training. All this being evidence of our hypothesis.

Our insights on weight oscillations and their role in quantization robustness open new horizons for model quantization approaches. Our regularization method especially creates interesting possibilities for cross-bit robustness, potentially making our regularization method more appealing than QAT when the goal is to deploy or relase a single set of weights that works across different bit widths or maybe even quantizers. While the regularizer used in our experiments should be viewed as an initial step, we expect that quantization robustness could be further improved by developing oscillation-inducing methods that are adaptive to different learning rates, layer statistics or phases of the training process.

## Broader Impact

This paper presents work whose goal is to advance the field of Machine Learning. There are many potential societal consequences of our work, none of which we feel must be specifically highlighted here.

**Acknowledgments:** Authors like to thank many funding agencies and colleagues for useful discussions.

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

## A. Appendix

### A.1. 2-layer with single weights

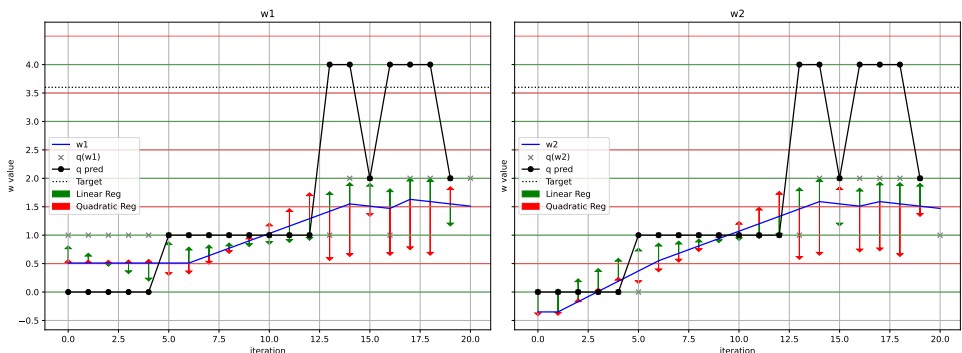

*Figure 4.* We repeat the toy model experiments, but this time with 2 weights, taking into account that the linear term is no longer 0 in the gradient. We notice at epoch 15 and 18 where the prediction of the quantized model is greater than y, the effect of the terms flip for $w_2$.

Consider a linear model $f(x) = w_2 w_1 x$, with $w_1, w_2$, input $x$, and target $y \in \mathbb{R}$. The quantized version of this model is defined as $f_q(x) = q(w_2)q(w_1)x$, where $q(\cdot)$ is the quantizer from Eq. 1. The quadratic loss for the model is given by

$$\mathcal{L}(f(x)) = \frac{1}{2}\big(w_2 w_1 x - y\big)^2$$

The difference compared to full-precision optimization is then given as

$$\delta_{\mathcal{L}} = \mathcal{L}(f_q(x)) - \mathcal{L}(f(x)) \tag{25}$$

$$= \frac{1}{2}\Big[\big(q(w_2)q(w_1)x - y\big)^2 - \big(w_2 w_1 x - y\big)^2\Big] \tag{26}$$

$$= \frac{1}{2}\Big[\big(q(w_2)q(w_1)x\big)^2 - \big(w_2 w_1 x\big)^2 - 2y\big(q(w_2)q(w_1)x - w_2 w_1 x\big)\Big] \tag{27}$$

$$= \frac{1}{2}x^2\Big[q(w_2)^2 q(w_1)^2 - w_2^2 w_1^2\Big] + yx\Big[w_2 w_1 - q(w_2)q(w_1)\Big] \tag{28}$$

The loss difference decomposes into:

$$\underbrace{\frac{1}{2}x^2\Big(q(w_2)^2 q(w_1)^2 - w_2^2 w_1^2\Big)}_{\text{quadratic term}} \quad + \quad \underbrace{yx\Big(w_2 w_1 - q(w_2)q(w_1)\Big)}_{\text{linear term}}$$

Taking the derivative of $\mathcal{L}$ with respect to $w_1$:

$$\frac{\partial \delta_{\mathcal{L}}}{\partial w_1} = \frac{\partial}{\partial w_1}\Big(\mathcal{L}(f_q(x)) - \mathcal{L}(f(x))\Big) \tag{29}$$

$$= \frac{\partial}{\partial w_1}\left[\frac{1}{2}x^2\Big(q(w_2)^2 q(w_1)^2 - w_2^2 w_1^2\Big) + yx\Big(w_2 w_1 - q(w_2)q(w_1)\Big)\right] \tag{30}$$

$$= x^2\left[q(w_2)^2 q(w_1)\frac{\partial q(w_1)}{\partial w_1} - w_2^2 w_1\right] + yx\left[w_2 - q(w_2)\frac{\partial q(w_1)}{\partial w_1}\right] \tag{31}$$

Using the STE approximation from Eq. 4, we get:

$$\frac{\partial \hat{\delta}_{\mathcal{L}}}{\partial w_1} = x^2\left[q(w_2)^2 q(w_1) - w_2^2 w_1\right] + yx\left[w_2 - q(w_2)\right] \tag{32}$$

We note that the linear term is no longer zero in the gradient and thus for a model consisting of 2 single weight layers we see that there is additional effects from QAT other than oscillations. Additionally because of the non-linearity of the rounding operation, even with the absence of a non-linear activation function, we can no longer reduce the model to a single weight.

## A.2. Hyperparameters

### A.2.1. RESNET-18

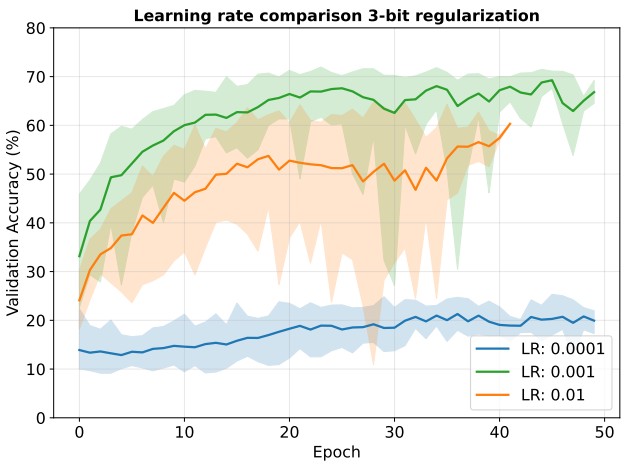

| $\lambda$ | 3-bit (%) | Ternary (%) |
|------|-----------------|-----------------|
| 0.25 | $68.77 \pm 0.19$ | $47.85 \pm 5.51$ |
| 0.50 | $69.47 \pm 1.11$ | $46.77 \pm 4.83$ |
| 0.75 | $70.08 \pm 0.40$ | $46.86 \pm 3.01$ |
| 1.00 | $66.20 \pm 4.05$ | $47.33 \pm 2.06$ |
| 1.25 | $69.31 \pm 0.32$ | $43.14 \pm 6.62$ |
| 1.50 | $68.96 \pm 0.30$ | $46.73 \pm 3.91$ |
| 1.75 | $69.92 \pm 0.11$ | $47.02 \pm 4.19$ |

*Figure 5.* Mean over 3 runs of the best validation accuracy for different lambdas. Training a ResNet-18 from scratch. Both ternary and 3-bit is at $10^{-3}$ learning rate and 50% of the data used for training. The plot shows three learning rates, where we for each learning ratue evaluate with the $\lambda$s in the rhs. table. The colored background covers the range between the maximum and minimum value of the quantized validation accuracy with the given $\lambda$s.

In Fig. 5 we see the results of a hyperparameter search over different learning rates and $\lambda$s for a ResNet-18 model. There is a clear trend of seeing the best performance at a learning rate of $10^{-3}$. We note that interestingly there is a comparable performance for a wide range of $\lambda$s, indicating that it is the presence of oscillations which is important for quantization robustness, and not the exact frequency of oscillations.

### A.2.2. TINY VIT

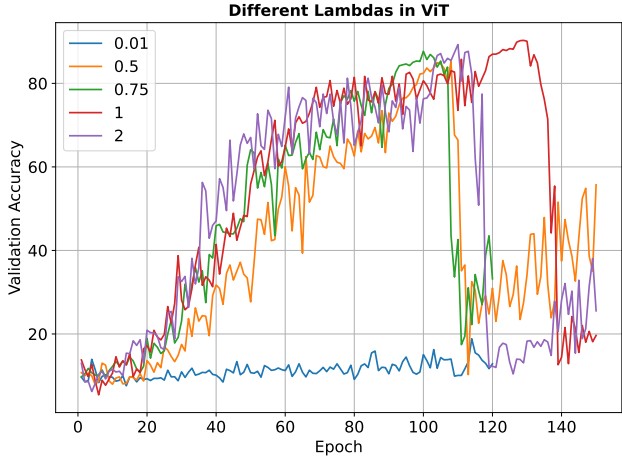

| $\lambda$ | 3-bit (%) | Ternary (%) |
|------|-----------|-------------|
| 0.01 | 18.85 | - |
| 0.5 | 85.21 | 15.10 |
| 0.75 | 87.68 | - |
| 1.0 | 90.29 | 13.04 |
| 2.0 | 89.31 | 14.16 |
| 2.5 | - | 13.70 |
| 5.0 | - | 14.20 |

*Figure 6.* Validation accuracy at different $\lambda$ values and the corresponding best validation accuracies for 3-bit and 2-bit configurations for a single run. Learning rate is set to 1e-4 for fine-tuning. For the 2-bit we test higher $\lambda$ but still see no improvemenet in accuracy. We note how all the $\lambda$s lies close to each other, except for the low of $10^{-2}$

Fig. 6 We note how also the Tiny Vit seems to allow for a wide range of $\lambda$s even though we this time note that $\lambda = 1$ performs significantly better than the others.

### A.3. Epochs and cross-bit robustness

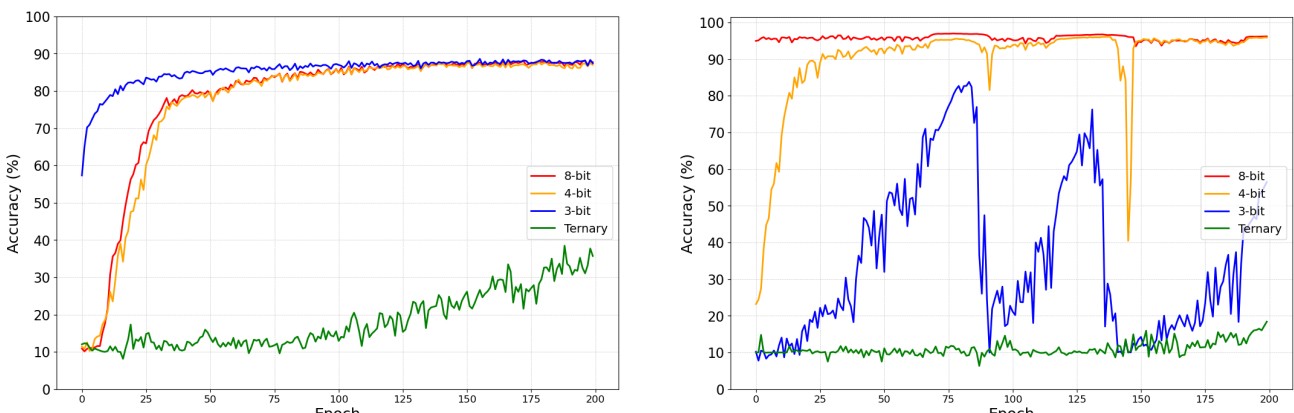

*Figure 7.* Left is the validation accuracy during training of a ViT with QAT at different bits, right is for our regularization. Both QAT and regularization is trained with a 3-bit quantizer. We note how the order of convergences for cross-bit changes between QAT and our model and that QAT cross-bit robustness especially depends on number of epochs trained.

There is an interesting interaction between number of epochs trained and robustness both of our method and QAT. We note how QAT converges first for the target-bit and then over time also converges for the 4 and 8-bit. Additionally we see that QAT seems upper-bounded by the target-bit performance, while this is not the case for our metho. Fig. 7,

### A.4. Convergence behaviour of Tiny ViT

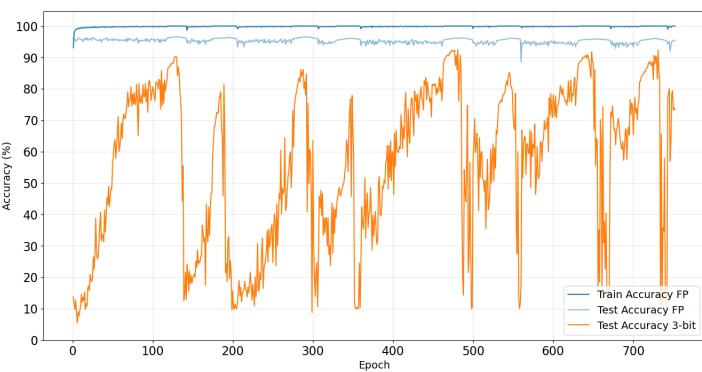

*Figure 8.* Regularization with a 3-bit quantizer on a Tiny ViT. We note the peculiar behaviour of the orange line, which is the validation accuracy on the target-bit performance. The performances cycles between $\approx 90\%$ and $10\%$, while the full precision accuracy (The model evaluated without quantized weights) stays some-what stable.

Fig. 8 shows the convergence behaviour of the full precision weights and the quantized weights at target-bit. We note how the Tiny ViT displays a peculiar convergence behaviour, where the accuracy will break, only to go up again. In the Tiny Vit model we quantize the self-attention layers, it is already noted in (Liu et al., 2023) that ViTs are especially vulnerable to quantization of the query and key of a self-attention layer, which might be related to the convergence behaviour we see.

## A.5. Counting oscillations

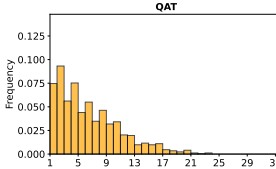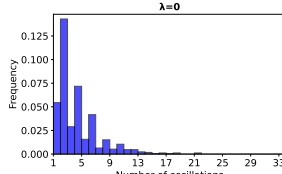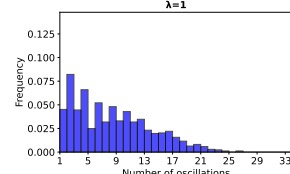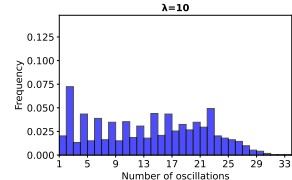

*Figure 9.* The plots shows the weights with a total oscillation count $> 0$ in a baseline model, a QAT model and a model regularized with $\lambda = 1$ respectively. The y-axis is the percentage of the total weights in the first convolutional layer in a ResNet-18 trained from scratch for 50 epochs. For the baseline model we log the full precision weights at each epoch and then apply the quantizer after training, where as for the regularized model we simply log both the full precision and quantized weights at each epoch.

Using the counting method proposed by (Nagel et al., 2022), we notice the biased odd distribution of the bins. In the plot used for the main paper, we count each bin the histogram in iterations of 2. Additionally we note that the proposed method misses the first oscillation count in each weight history.

