# OpenReview forum: "Oscillations Make Neural Networks Robust to Quantization"
_ICML.cc/2025/Conference — Submitted to ICML 2025_

### Official Review · Reviewer_rX35 · 2025-02-17

**Overall Recommendation:** 3

**Summary:**

The paper investigates the role of oscillations in Quantization Aware Training (QAT) for neural networks. Traditionally, oscillations in QAT, caused by the Straight-Through Estimator (STE), are considered undesirable artifacts. However, this paper presents a different perspective by proposing that these oscillations can actually improve the robustness of neural networks to quantization.

## update after rebuttal
Thank you for your response and the insightful experiments. As we enter the era of multimodal LLMs, quantization across diverse domains is gaining traction—recent PTQ and QAT studies increasingly incorporate evaluations spanning multiple modalities. Exploring these aspects would enhance the relevance and impact of your contributions.  Also, I suggest that the authors consider using a model such as Qwen-2.5-0.5B to demonstrate the proof of concept. As a result, I will keep my original score.

**Claims And Evidence:**

The paper's claim is clearly supported by evidence and mathematical equations to substantiate its validity.

**Essential References Not Discussed:**

The author could provide a more detailed discussion of related work in QAT, such as Outlier Suppression [Wei'22], Outlier Suppression+ [Wei'23], BiE [Zou'24], EfficientDM [He'23], FP8 Quantization [Kuzmin'22], and PackQViT [Dong'23].

**Experimental Designs Or Analyses:**

- Is it feasible to apply transformer-based models from other domains, such as NLP (BERT, OPT-125m) and time series (PatchTST, StanHop)?
- What is the quantization performance for weight and activation function quantization?
- What is the quantization performance with different quantization lambda values?

**Methods And Evaluation Criteria:**

The paper uses mathematical equations to explain the existence of weight oscillations during QAT and how regularization can induce them. For the experiment, accuracy is used to demonstrate that the quantization robustness makes sense.

**Other Comments Or Suggestions:**

Place the table caption above the table.

**Other Strengths And Weaknesses:**

The mathmatical expressions to prove why the mechanism that leads to weight oscilla-
tions during QAT and why regularize is generally easy understand to me.

**Questions For Authors:**

Since some authors claim that Quantizable Transformer [Bondarenko'23] is a QAT method, I have a question. Can you help me explain how the weight oscillations are related to the influence of outliers in Quantizable Transformer [Bondarenko'23] and OutEffHop [Hu'24]? The clip-softmax in [Bondarenko'23] and Softmax_1 in [Hu'24] seem to function as a form of regularization—is that correct? If so, how does your method relate to these approaches?

**Relation To Broader Scientific Literature:**

It offers a more resource-efficient quantization method that delivers similar performance to QAT.

**Theoretical Claims:**

The paper not include the theoretical claims

---

> ### Author Rebuttal · Authors · 2025-03-31
>
> Thank you for the careful reading of our manuscript, your helpful comments and suggestions of relevant literature.
>
> * Thank you for the suggestion on exploring the quantization performance for different $\lambda$ values. We have now expanded the analysis in A.2 to span a larger range of $\lambda$. In the final version, we will provide additional data regarding the quantization performance for different $\lambda$ values. Also see the response to reviewer tzVB for additional discussions. We present an ASCII rendering of these experiments below:
>
> ```Accuracy vs Regularization Lambda
>
>  100 |
>   90 |                      @     @      @
>   80 |               @                         @
>   70 |          @
>   60 |
>   50 |                                               @
>   40 |
>   30 |
>   20 |
>   10 |                                                    @
>    0 |__________________________________________________________
>       10^-3      10^-2      10^-1      10^0      10^1      10^2
>                              λ (log scale)
>
> ```
>
> * Thank you for providing an extensive list of additional related work, and specifically the questions related to Bondarenko et al. and Hu et al. Initial reading of these two papers do not indicate that they are focused on the specific effects of oscillations in QAT that we are concerned with. For instance, Bondarenko et al. (2023) is primarily concerned with _activation quantization_ during _post-training quantization (PTQ)_. Similarly, Hu et al. (2024) is concerned with handling outliers efficiently which they also argue has an effect on improving the PTQ performance.
> We will take a closer look at suggested references, and make sure to cite all relevant papers in the final version.
>
> * Although in this paper we focus on a computer vision tasks to demonstrate the effect of oscillations, we believe that the results would transfer to other domains.
> * The theoretical analysis presented in this work explicitly focused on weight quantization, for which we provide experimental performance data in Tables 1-3. We do appreciate the reviewer's suggestion on attempting this for quantizing activation maps. This would be exciting future work as it would entail expanding the theoretical analysis to include activation maps, and performing additional experiments specifically to analyze the quantization effects on activation maps.
>
> Bondarenko et al. "Quantizable Transformers: Removing Outliers by Helping Attention Heads Do Nothing" (2023)
>
> Hu et al. "Outlier-Efficient Hopfield Layers for Large Transformer-Based Models" (2024)

---

> > ### Comment · Reviewer_rX35 · 2025-04-01
> >
> > Thank you for your response and the insightful experiments. I believe your work would be significantly strengthened by extending the analysis to both weight and activation quantization. As we enter the era of multimodal LLMs, quantization across diverse domains is gaining traction—recent PTQ and QAT studies increasingly incorporate evaluations spanning multiple modalities. Exploring these aspects would enhance the relevance and impact of your contributions. As a result, I will keep my original score.

---

> > > ### Author Response · Authors · 2025-04-08
> > >
> > > We thank the reviewer for their response to our rebuttal. We would like to clarify further the two points you mention in your response.
> > >
> > > 1) **Activation quantization**
> > >
> > > While this paper focuses on weight quantization and the positive role of QAT oscillations, we believe that activation quantization represents an interesting but _orthogonal_ direction. The insights from our current weight analysis and experiments stand clearly on their own and the absence of an activation quantization analysis does not diminish the novelty or significance of our current contributions.
> > >
> > > Further, given that weight quantization remains one of the most common and practical quantization scenarios - particularly for reducing memory requirements during fine-tuning and/or inference - we would argue that our contributions are highly relevant to current practices and researchers.
> > >
> > > Indeed, some of the most prevalent quantization methods (e.g. AWQ[1], GPTQ[2], QLoRA[3]) focus solely on quantization of the weights, due to the fact that memory requirements imposed by parameter count is one of the main bottlenecks when dealing with transformer-based models. Thus our study of weight oscillations is directly relevant to this prominent class of methods.
> > >
> > > So given the widespread usage of weight quantization and our novel contributions wrt. to QAT - we argue that the absence of an analysis of activation quantization does not diminish the current manuscripts relevance or significance.
> > >
> > > [1] AWQ: Lin, Ji, et al. "AWQ: Activation-aware weight quantization for on-device llm compression and acceleration." 2024
> > >
> > > [2] Frantar, Elias, et al. "GPTQ: Accurate post-training quantization for generative pre-trained transformers." (2022).
> > >
> > > [3] QLORA: Efficient Finetuning of Quantized LLMs, 2023
> > >
> > >
> > >
> > > 2) **Multimodal/quantization of LLMs**
> > >
> > > We appreciate the reviewers emphasis on broad applicability. In our current analysis and experiments we already include transformer-based models (specifically ViT). However, expanding to full-scale LLMs  involves substantial computational demands particularly when analyzing over a range of bits and retraining dynamics. As such, we view this as an interesting but out-of-scope direction for this paper.
> > >
> > > That said we emphasize that our theoretical and empirical results around oscillations, and the conditions under which they enhance quantization robustness, are general in nature. Given that we already shown their effectiveness on transformers in the CV setting, we anticipate they will translate to LLMs and multimodal settings as well.
> > >
> > >
> > > Given these further clarifications, we respectfully invite the reviewer to reconsider whether we have sufficiently addressed their initial concerns. In any case, we are grateful for your feedback and for supporting our work.

---

### Official Review · Reviewer_3CFm · 2025-03-09

**Overall Recommendation:** 2

**Summary:**

This work researches the oscillation effect during quantization-aware training (QAT) from a novel perspective. While most previous work identifies oscillation as a negative effect and tries to minimize it during QAT, the author of this work focuses more on the beneficial influence of preserving model performance. Based on a theoretical analysis of a linear model with a single weight, the author unveils that the dynamic (gradient of STE) leads to clustering around quantization thresholds. To use this clue, this work introduces a regularization term (named (OsciQuant) to emulate the effects. Experiments on MLP/ResNet/ViTs across various datasets (CIFAR-10/ImageNet-1K) demonstrate the effectiveness of the OsciQuant under some specific settings.

**Claims And Evidence:**

Most of the claims are supported by theoretical or empirical analysis. However, there are still some concerns:

- Most "theoretical" analyses in section 4 are straightforward and do not provide much insight. STE will lead all quantized weights to clustering around quantization thresholds, not limited to the weights that are oscillating.
- The author proposed the regularization term in Eq. 23 simply from the point that "... we let the regularization term be similar to the
quadratic term in Eq. 14". This is not convincing as the design for regularization can be highly diverse. What should be the weighting coefficient instead of $\frac{\lambda}{2}$? Why weight magnitude $x$ or $x^2$ is not included in $R_{\lambda}(w)$?

**Essential References Not Discussed:**

Previous work [1] also proposed a regularization term to suppress the oscillation during QAT of transformer-based models but it is not discussed in this work. In addition, [2] discusses the oscillation of PTQ, which should also be included in the related work.

[1] Quantization Variation: A New Perspective on Training Transformers with Low-Bit Precision, TMLR 2023

[2] Solving Oscillation Problem in Post-Training Quantization Through a Theoretical Perspective, CVPR 2023

**Experimental Designs Or Analyses:**

I have checked the soundness/validity of all experimental designs. However, I think the results across all experiments show that OsciQuant only outperforms QAT under specific settings. "Comparable performance" with QAT does not lead to the conclusion that "weight oscillations are
a necessary part of the QAT training and should be preserved"

**Methods And Evaluation Criteria:**

The proposed methods and evaluation make sense to me. QAT on more tasks such as object detections or even NLP could be considered in future work.

**Other Comments Or Suggestions:**

Please cite the correct version of some previous work instead of the arxiv version. For example: "Additive powers-of-two quantization: An efficient non-uniform discretization for neural networks" should be cited as ICLR paper

    @inproceedings{
    Li2020Additive,
    title={Additive Powers-of-Two Quantization: An Efficient Non-uniform Discretization for Neural Networks},
    author={Yuhang Li and Xin Dong and Wei Wang},
    booktitle={International Conference on Learning Representations},
    year={2020},
    url={https://openreview.net/forum?id=BkgXT24tDS}
    }

**Other Strengths And Weaknesses:**

N/A

**Questions For Authors:**

Please see all the comments above.

**Relation To Broader Scientific Literature:**

The "oscillation" phenomenon in QAT has already been discussed in many previous works but this work is the first to focus on the beneficial effect of oscillation. The author did not overclaim their contribution and the majority of previous literature is cited.

**Theoretical Claims:**

I have checked the correctness of all proofs for theoretical claims in this work. One problem is that in section 3.2, $\frac{\hat{\partial} q}{\hat{\partial} w} = 1$ only holds for the $w$ falls in the quantization region in STE. This is not highlighted and is directly used in Eq.13-Eq.14 and Eq. 24.

---

> ### Author Rebuttal · Authors · 2025-03-31
>
> Thank you for the careful reading of our manuscript, your helpful comments and pointers to relevant literature.
>
> We agree that the theoretical analyses in Section 4 are straightforward. At the same time, they provide essential intuition and motivation for the empirical results because they give an explicit description of the mechanism behind oscillations and clustering in a simplified setting.
>
> You are completely right that in our model, there can in principle be clustering without oscillation and oscillation without clustering. We will add the following text at the end of Section 4 to clarify this point:
>
> "In this model, the clustering and/or oscillation of individual weights depends on the relative influence of L(w) and $\delta L$".
>
> We will also elaborate on the choice of the regularizer, adding the following in the paragraph following (23):
> "In this term, we replaced the factor $x^2$ by a hyperparameter $\lambda$, since the precise expression of $x^2$ is specific to the model studied in Section 3 (see also Appendix A). We empirically find that this regularizer is sufficient to induce oscillations and show their positive effect. The exploration of the design space of oscillation-inducing regularizers, including layer-dependent and/or adaptive scale factors, is left to future work."
>
> **Oscillations and QAT**: Regarding your observation on the necessity of oscillations, we would like to point out that the fact that oscillations are necessary follows from previously published experiments, as described in the first paragraph of Section 7, and we do not claim to make a novel contribution in this regard.
>
> Additionally, regarding the sufficiency of oscillations, our results are empirical and therefore of course cannot unambiguously show that oscillations are sufficient for QAT in all cases. However, as previous work has essentially claimed that oscillations are harmful, we believe that even showing experimentally that they are sufficient (using MLP, ResNet and Transformer) is a significant contribution, especially when combined with previous results on them being necessary at least for part of the training process.
>
> In one of the places in the initial manuscript (L87) we believe the claims relating oscillations and QAT is strong and we will adjust to "... our results suggest that weight oscillations capture many of the beneficial effects of QAT ..." instead of "all the beneficial effects of QAT".
>
> **Theoretical claim on STE and region of quantization**: As we note in lines 155-158, the scale factor is chosen to cover the range of the tensor to be quantized, so there is no clamping operation in our quantizer defined in Equation (1). This means that all weights will always be within the quantization region and subsequently that the STE is always 1. We will further clarify this in line 191-192 "Using the STE and recalling that the STE gradient simplifies to $\frac{\hat{\partial}q}{\hat{\partial}w}=1$ (note that there is no clamping in our setting, see Equation (2)), ..."
>
> We will cite the correct version of "Additive powers-of-two quantization: An efficient non-uniform discretization for neural networks" and double check that we are citing the correct version of our other references.
>
> We thank the reviewer for suggesting [1,2]. We will include [1] in the background section on weight oscillations. The type of oscillation discussed in [2] refers to fluctuations in the reconstruction loss across network layers during PTQ, which is traced back to differences in module capacity between adjacent layers. This is different to our area of investigation, oscillations during QAT, which is a periodic change in the quantized value of the weights. We will mention this in the background.
>
> [1] Quantization Variation: A New Perspective on Training Transformers with Low-Bit Precision, TMLR 2023
>
> [2] Solving Oscillation Problem in Post-Training Quantization Through a Theoretical Perspective, CVPR 2023

---

### Official Review · Reviewer_tzVB · 2025-03-10

**Overall Recommendation:** 4

**Summary:**

The paper uses a linear model to explain the mechanism of weight oscillation during quantization-aware training (QAT). It discovers that the oscillation is because the loss function with quantized weights encourages the latent weights to cluster around the edge of quantization buckets, not the center. The paper then proposes to add a regularization term to the original loss function, i.e., $\frac{1}{N} \sum w_q^2 - w^2$, that has the same effect of weight oscillation to replace QAT. Experiment on toys models such MLP-5 on CIFAR-10 and finetuned ImageNet models show that this regularization term (OsciQuant) can lead to a similar accuracy as QAT.

**Claims And Evidence:**

Please refer to the strengths and weaknesses section.

**Essential References Not Discussed:**

Not that the reviewer is aware of.

**Experimental Designs Or Analyses:**

Please refer to the strengths and weaknesses section.

**Methods And Evaluation Criteria:**

Please refer to the strengths and weaknesses section.

**Other Comments Or Suggestions:**

Please refer to the strengths and weaknesses section.

**Other Strengths And Weaknesses:**

The paper presents a very interesting view from the gradient of loss diff $L(q(w)) - L(w)$. The analysis of QAT's encouraging latent weights to be clustered around bucket edge is already interesting enough to be shown to the public.

However, the regularizer does not seem to fully isolate the benefit of QAT because: (1) if moving latent weights to bucket center is all what QAT does, the lamda ablation in the table beside Figure 5 (in Appendix A.2) should show that the accuracy will become increasing better when lambda gets bigger, which is not the case; (2) Figure 7 shows that QAT apparently converges much smoother without accuracy dips (loss spikes). The absolute value of $L(q(w))$ might play a role there.

It would be interesting to see how well OsciQuant does when lambda is extremely large, i.e., all latent weights are nearly at the bucket edge, and the tie is broken randomly after training. The pattern of lambda is still not clear from the table on line 616 (please make a label for the table).

**Questions For Authors:**

Please refer to the strengths and weaknesses section.

**Relation To Broader Scientific Literature:**

The contribution is related to quantization-aware training.

**Theoretical Claims:**

Checked equation 1 to 24.

---

> ### Author Rebuttal · Authors · 2025-03-31
>
> Thank you for the careful reading of our manuscript and your insightful comments, and for this characterisation of our work: "The analysis of QAT's encouraging latent weights to be clustered around bucket edge is already interesting enough to be shown to the public."
>
> Regarding point (1), we agree that the accuracy should become better as $\lambda$ gets bigger, and eventually deteriorate as the regularizer starts to dominate the loss for large values of $\lambda$. This is indeed the case if we vary $\lambda$ in a wider range. We performed additional experiments and observe this trend. We will add the corresponding figure to the final version. Please find an ASCII rendering of the data below:
>
> ```Accuracy vs Regularization Lambda
>
>  100 |
>   90 |                      @     @      @
>   80 |               @                         @
>   70 |          @
>   60 |
>   50 |                                               @
>   40 |
>   30 |
>   20 |
>   10 |                                                    @
>    0 |__________________________________________________________
>       10^-3      10^-2      10^-1      10^0      10^1      10^2
>                              λ (log scale)
>
> ```
>
> We also agree with your point (2) that oscillations do not fully reproduce the training dynamics of QAT. While we do discuss this already in lines 406-421, we will further clarify this in the manuscript by adding the following text after line 421:
> "Preliminary observations indicate that some of the secondary effects of QAT can be beneficial for training dynamics and convergence, see Appendix A.4".
>
>
> **Training dynamics of ViT with OsciQuant**: We currently do not have a clear explanation for the behaviour reported in Figures 7 and 8 for ViT. The closest hypothesis we have is the attribution we already make in A.4 using the arguments from Liu et al. 2023, who point out that "the interdependence between quantized weights in query and key of a self-attention layer makes ViT vulnerable to oscillation" which might explain the behaviour observed when we only induce oscillations using our method.
>
> We have now analyzed the training curves for other models like ResNet-18. We do not observe these large drops in accuracy for ResNet-18, which leads us to believe this is an artifact of transformer-based models as suggested by Liu et al. 2023.
>
> Liu et al. "Oscillation-free Quantization for Low-bit Vision Transformers" 2023.

---

> > ### Comment · Reviewer_tzVB · 2025-04-08
> >
> > I read the authors' and other reviewers' comments. I agree a lot with reviewer 3CFm that (1) Most "theoretical" analyses in section 4 are straightforward; (2) "comparable performance with QAT does not lead to the conclusion that weight oscillations are a necessary part of the QAT training and should be preserved".
> >
> > However, I think that the value of this paper is not that OsciQuant outperforms QAT with an alternative regularizer, but rather that the discussion on oscillation does potentially bring us closer to the underlying pattern of what QAT is doing. Straightforward analysis is a plus for me. I will therefore provide the support.
> >
> > (Side comment: the ASCII figure looks very nice.)

---

### Official Review · Reviewer_f7Hw · 2025-03-11

**Overall Recommendation:** 2

**Summary:**

This paper challenges the traditional view of oscillations in QAT as undesirable, arguing they can enhance robustness. Through theoretical analysis of linear models, the authors decompose the QAT loss gradient into the original full-precision component and an oscillation-inducing term. Then they introduced OsciQuant, a novel regularization method that intentionally encourages oscillations, contrary to conventional QAT approaches. This method leverages oscillatory behavior to mitigate quantization effects, improving cross-bit robustness. Experimental results on ResNet-18 and Tiny ViT demonstrate that OsciQuant matches or surpasses QAT performance at 3-bit weight quantization, while maintaining high accuracy at other bit-widths, proving its effectiveness in preserving model performance under quantization.

## update after rebuttal
I have carefully reviewed the rebuttal and considered the opinions of the other reviewers. I am inclined to maintain my original score. While I acknowledge that the exploration of oscillation effects in quantization presented in this paper offers some interesting insights, I agree with Reviewer 3CFm that the overall contribution is somewhat straightforward. In its current form, the paper does not meet the bar of novelty and technical depth expected for ICML.

**Claims And Evidence:**

The main claim of the paper is that weight oscillations during training are beneficial for quantization robustness. The authors provide both theoretical and empirical evidence to support this claim.

Theoretically, they analyze a simple linear model and show that the gradient of the loss function can be decomposed into two terms: the original full-precision loss and a term that causes quantization oscillations. And this mechanism causes weights to move towards the quantization thresholds.

Based on this observation, they develop a regularization method that encourages weight oscillations during training. Empirically, they evaluate their method on ResNet-18 and Tiny ViT, and show that it can match QAT accuracy at >3-bit weight quantization.

**Essential References Not Discussed:**

N/A

**Experimental Designs Or Analyses:**

The experimental designs and analyses are sound and convincing. The authors evaluate their method on CIFAR-10 benchmark datasets, and show that it can achieve competitive accuracy compared to QAT. They also evaluate the robustness of their method to different levels of quantization, and show that it can maintain close to full precision accuracy even at low bit widths.

**Methods And Evaluation Criteria:**

Yes. Though the evaluation is limited to CV task and CIFAR-10 dataset only. Extensive experiments on other tasks and datasets would further enhance the findings of the paper.

**Other Comments Or Suggestions:**

N/A

**Other Strengths And Weaknesses:**

The paper is well-written and has good theoretical proof. But the contribution, significance and novelty are limited. Thus, I believe it's slightly below the acceptance line.

**Questions For Authors:**

N/A

**Relation To Broader Scientific Literature:**

The paper builds upon previous research on quantization-aware training, quantization error minimization, and oscillations in QAT.

**Theoretical Claims:**

Yes, see Claims and Evidence.

---

> ### Author Rebuttal · Authors · 2025-03-31
>
> Thank you for your thorough and careful reading of our manuscript.
>
> We are pleased to read that you found our paper to be "well-written and has good theoretical proof" and about your assessment that the "experimental designs and analyses are sound and convincing".
>
> However, we respectfully disagree with the statement that "the contribution, significance and novelty are limited".
>
> Our paper demonstrates for the first time the beneficial effect of oscillations in quantization-aware training (QAT), which challenges a major belief on the mechanisms underlying QAT. We believe our contribution to be both novel, since it is the first to show the beneficial effect of oscillations, and significant, since we expect that our result will significantly impact the thinking and design principles behind the development of future QAT and quantization-aware methods in general, which has traditionally emphasized aligning weights to bucket centers.
>
> We would also like to point out that other reviewers commented positively on the novelty and significance, with Reviewer tzVB writing, "The analysis of QAT's encouraging latent weights to be clustered around bucket edge is already interesting enough to be shown to the public" and Reviewer 3CFm confirming the novelty of our approach in writing "this work is the first to focus on the beneficial effect of oscillation."

---

### Decision · Program_Chairs · 2025-05-01

**Decision:**

Reject

**Comment:**

Summary: This paper challenges the notion that oscillations in QAT are harmful, showing through linear model analysis that they can enhance robustness by encouraging weights to cluster near quantization thresholds. It introduces OsciQuant, a regularization method that promotes beneficial oscillations, achieving QAT-level or better accuracy at low bit-widths across MLPs, ResNets, and ViTs.

Review summary: The reviewers appreciate the writing and the study of oscillations, however, one reviewer points out that this has been studied in previous works. The regularization proposed is here is novel, but overall, the authors generally found the contributions to be insufficiently novel to merit acceptance.

Recommendation & justification: I recommend rejection. While the paper does provide an interesting study about oscillations in QAT, the overall contributions are not strong enough for acceptance.